# Water masses shape pico-nano eukaryotic communities of the Weddell Sea

Olga Flegontova[1,2], Pavel Flegontov[1,2], Nikola Jachníková[3], Julius Lukeš [1,3] & Aleš Horák [1,3✉]

Polar oceans belong to the most productive and rapidly changing environments, yet our understanding of this fragile ecosystem remains limited. Here we present an analysis of a unique set of DNA metabarcoding samples from the western Weddell Sea sampled throughout the whole water column and across five water masses with different characteristics and different origin. We focus on factors affecting the distribution of planktonic pico-nano eukaryotes and observe an ecological succession of eukaryotic communities as the water masses move away from the surface and as oxygen becomes depleted with time. At the beginning of this succession, in the photic zone, algae, bacteriovores, and predators of small eukaryotes dominate the community, while another community develops as the water sinks deeper, mostly composed of parasitoids (syndinians), mesoplankton predators (radiolarians), and diplonemids. The strongly correlated distribution of syndinians and diplonemids along the depth and oxygen gradients suggests their close ecological link and moves us closer to understanding the biological role of the latter group in the ocean ecosystem.

[1] Institute of Parasitology, Biology Centre, Czech Academy of Sciences, České Budějovice, Czech Republic. [2] Department of Biology and Ecology, Faculty of Science, University of Ostrava, Ostrava, Czech Republic. [3] Department of Molecular Biology, Faculty of Science, University of South Bohemia, České Budějovice, Czech Republic. ✉email: ogar@paru.cas.cz

Heterotrophic protists are a vital component of the ocean's plankton throughout the water column[1–3]. Even in the photic layer, they are more diverse and abundant than primary eukaryotic producers[4]. Their distribution is mainly influenced by a combination of abiotic factors (with temperature and oxygen concentration being the most important) and biotic interactions[5]. Marine protist communities are now actively mapped by large metabarcoding projects[4–12], but those are focused mostly on the tropical and temperate regions and on the sunlit ocean, where the bulk of the ocean's productivity takes place. Although polar oceans belong to one of the most productive and rapidly changing ecosystems on Earth[13], the Southern Ocean remains poorly represented in these large surveys of marine protists. With the exception of a single small-scale study that examined protists in several deep-ocean water masses[14], most studies of protists in the Southern Ocean focused on the photic layer[15–21]. Given the unique conditions during the polar winter, it is reasonable to assume that the heterotrophic lifestyle is of particular importance in the polar environment. The dark ocean, namely the mesopelagic (200–1000 m) and bathypelagic (1000–4000 m) layers that form by far the most voluminous biome on Earth, is generally sparsely covered by metabarcoding surveys[6,7,9,11]. To our knowledge, studies that investigate detailed depth stratification of communities across all relevant protist groups are rare, and such studies reporting samples from the Southern Ocean are non-existent. Therefore, we aimed to fill this gap.

The Weddell Sea hosts a well-known oceanographic feature, the Weddell Gyre. The formation/melting of sea-ice and melting of shelf ice create specific conditions that make this region the most important site for deep and bottom water formation for the whole Southern Hemisphere, and one of few such locations around the globe. This is also a crucial region of the ocean where gas exchange between the ocean and the atmosphere occurs, affecting oxygen and carbon dioxide levels in the deep ocean (below 200 m) much further north[22].

Below we provide an overview of the Weddell Sea water masses relevant to our work, drawing on information published elsewhere[22–25]. Surface Water (SuW) from depths of less than 100 m is influenced by sea-ice melt in the open water regions in summer and therefore it has low salinity, a cold temperature below 0 °C, and a high oxygen concentration due to gas exchange between the ocean and the atmosphere. At depths between 400 m and 1600 m is the Warm Deep Water (WDW), characterized by a temperature above 0 °C and a salinity above 34.6. WDW circulates clockwise within the Weddell Gyre and originates in the Circumpolar Deep Water (CDW) of the Antarctic Circumpolar Current which enters the Weddell Gyre at its eastern edge. CDW has a higher temperature and salinity compared to WDW and, in turn, originates in the North Atlantic Deep Water (NADW). Therefore, WDW has a long history of being in the deep ocean with no contact with the atmosphere and as a result has a low oxygen concentration. Sandwiched between SuW and WDW is Modified Warm Deep Water (MWDW), which results from the upwelling of WDW and mixing with SuW, so that the temperature, salinity, and oxygen concentration of the MWDW are intermediate between those of WDW and SuW.

Offshore the southern and western ice shelves of the Weddell Sea, extensive sea ice formation and brine precipitation in nearshore polynyas produce shelf water with high salinity (HSSW). HSSW (with the following properties: salinity of 34.60–34.85; surface-freezing temperature about −1.9 °C; oxygen-enriched) sinks along the continental slope and forms the Weddell Sea Bottom Water (WSBW, characterized by temperature under −0.7 °C and salinity above 34.6). When WSBW reaches the Weddell Gyre, mixing at the interface between the WSBW and WDW forms a new water mass, the Weddell Sea Deep Water (WSDW), characterized by a temperature between −0.7 °C and 0 °C and salinity above 34.63. The Weddell Gyre is an important site for the formation of the Antarctic Bottom Water (AABW), whose precursors are WSBW and WSDW. AABW is exported northward from the Weddell Gyre and constitutes a major ventilation component of the global abyssal ocean.

## Results

**Our samples in the context of Weddell Sea water masses.** Here we present a DNA metabarcoding study based on 110 marine planktonic samples collected on the western shelf of the Weddell Sea (eight locations) and in the Powell Basin (nine locations) (Fig. 1a, Suppl. Data 1). The samples were taken from depth transects with ~200 m increments; sampling depth ranged from 40 m to 3221 m (Fig. 1b). The samples collected in the Powell Basin belonged to four local water masses, defined by potential temperature, salinity, and oxygen concentration[22] as shown in Fig. 1c, d (see "Introduction" for abbreviations and details on the water mass origin). All 30 samples collected from the western shelf of the Weddell Sea from depths of 100–500 m (the maximum depth of the shelf was 534 m) do not correspond in their properties to any of the water masses described above, so we defined them simply as Shelf Water (ShW; Fig. 1e).

**Communities of pico-nano eukaryotes in the Weddell Sea evolve along with water masses.** To investigate the composition of planktonic eukaryotic (mainly protistan) communities in our depth transects, the V9 18S rRNA barcode was amplified and sequenced from DNA isolated from the pico-nano size fraction (0.8 to 20 µm). For details on DNA sequencing, read processing, operational taxonomic unit (OTU) definition, and their taxonomic annotation, see "Methods". After filtering out rare OTUs (with an abundance of 100 reads or less), we obtained 146.4 million reads (95.9% of the original reads) and 5048 eukaryotic OTUs (10.8% of the original OTUs). To investigate how eukaryotic pico-nano community composition depends on water masses, oxygen concentration and other environmental variables (depth, salinity, temperature, oxygen saturation, fluorescence intensity, latitude, longitude, and bottom depth at the sampling site), we (1) performed non-metric multidimensional scaling (NMDS) of Bray-Curtis distances between communities, (2) inferred ecological boundaries (places along environmental gradients where communities undergo a statistically significant change), and (3) used Mantel tests to estimated correlations of the community distance matrix with Bray-Curtis distance matrices based on individual environmental variables. Comprehensive results of these analyses are shown in Suppl. Fig. 1, and selected variables are presented in Fig. 2. The arrangement of communities in the two-dimensional NMDS space better corresponds to water masses than to depth; in other words, water masses subdivide the NMDS space into areas with smaller overlap compared to areas defined by ecological boundaries along the depth gradient (Fig. 2). It is noteworthy that the arrangement of communities in the NMDS space is corresponding no worse to oxygen concentration and salinity than to water masses (Fig. 2). Looking at the dataset in a different way (using the Mantel test), we find that oxygen concentration and depth are best correlated with the Bray-Curtis distances of the eukaryotic communities ($r = 0.83$ and 0.74, respectively) (Fig. 2, Suppl. Fig. 1). The distributions of oxygen concentration values passed the test for normality in five water masses, whereas the distributions of depth, salinity, and temperature failed this test in at least one water mass (Suppl. Fig. 2). Therefore, for the comparison of water masses with respect to oxygen concentration, we used the multiple ANOVA,

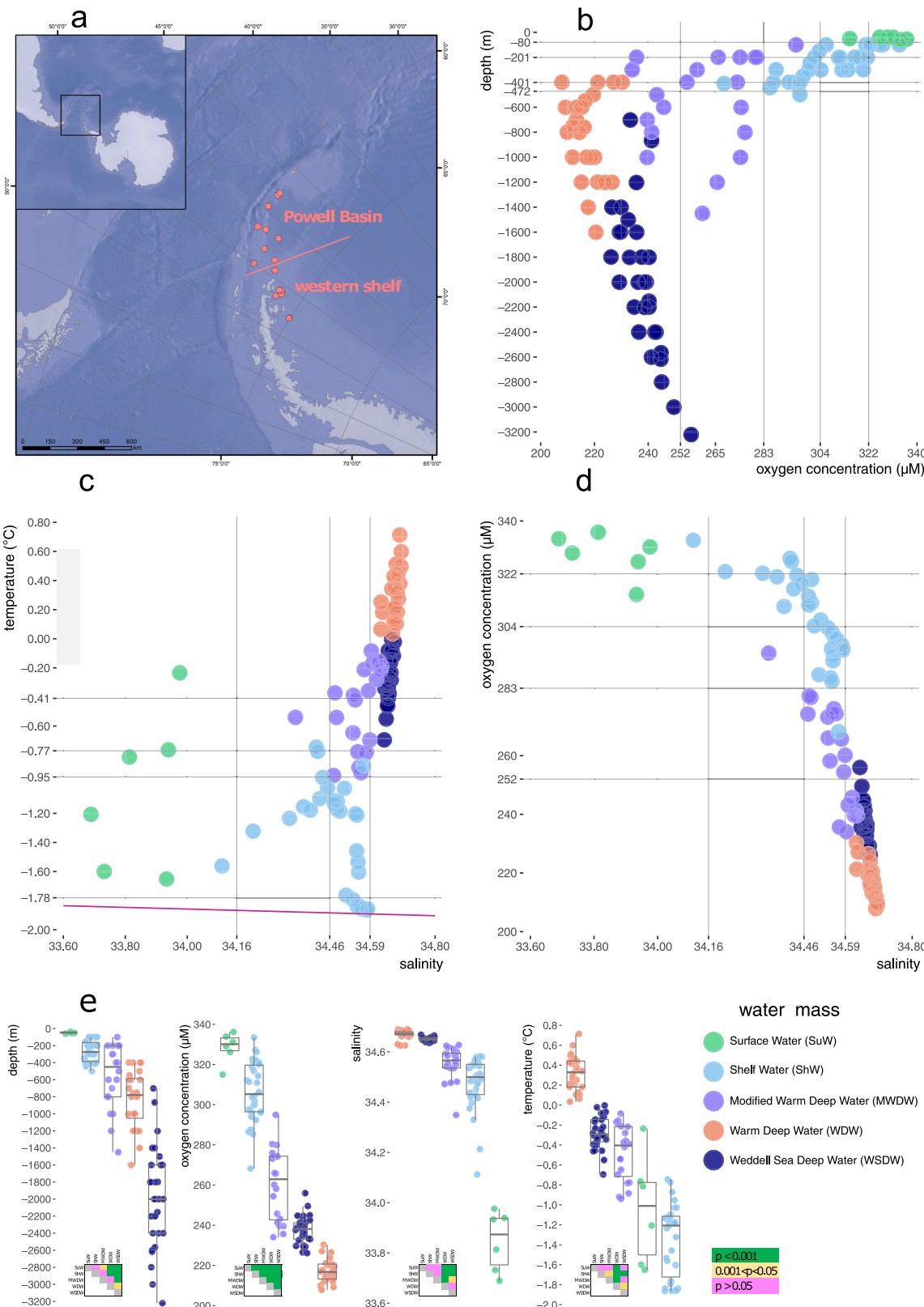

and for the comparison with respect to depth, salinity, and temperature, we used the non-parametric Kruskal–Wallis test followed by post-hoc Dunn's test. From the results of the ANOVA and Kruskal-Wallis analyses, as well as the boxplots, it is evident that oxygen concentration corresponds the best to the affiliation of the samples to the respective water masses in our dataset (Fig. 1e). We believe that this is not a coincidence, as oxygen concentration is known to depend on the time since the last contact of the water with the surface[22,26,27].

Next, we subdivided the eukaryotic pico-nano communities into clades and considered only those represented by at least 200,000 reads resulting in 26 taxonomic groups (Suppl. Data 3).

**Fig. 1 The geographical and environmental context of our metabarcoding samples.** A map of sampling locations is shown in panel (**a**), where the pink line separates samples collected on the Western Antarctic Shelf and those collected in the Powell Basin. The samples were attributed to water masses based on a salinity vs. temperature diagram (panel **c**, surface water freezing temperatures are shown with the magenta line), and five water masses sampled by us are color-coded according to the legend in the lower right corner. Sampling depths and the dependence of oxygen concentration on depth are shown in panel (**b**). Tight non-linear relationship of oxygen concentration and salinity is illustrated in panel (**d**). Distributions of these four key abiotic variables (depth, oxygen concentration, salinity, and temperature) across the five water masses sampled are shown in panel (**e**) in the form of boxplots, along with results of relevant dispersion analyses showing if water masses are significantly different according to a certain variable (multiple ANOVA in the case of oxygen concentration and the non-parametric Kruskal-Wallis test followed by post-hoc Dunn's test in the case of depth, salinity, and temperature). The statistical tests were applied to sample sets of the following sizes: 6 samples (Surface Water), 30 samples (Shelf Water), 20 samples (Modified Warm Deep Water), 24 samples (Warm Deep Water), 30 samples (Weddell Sea Deep Water). Ecological boundaries inferred for pico-nano eukaryotic communities along the depth, oxygen concentration, salinity, and temperature gradients are shown with vertical and horizontal gray lines in panels (**b**), (**c**), (**d**). The source data underlying this figure see in Suppl. Data 1.

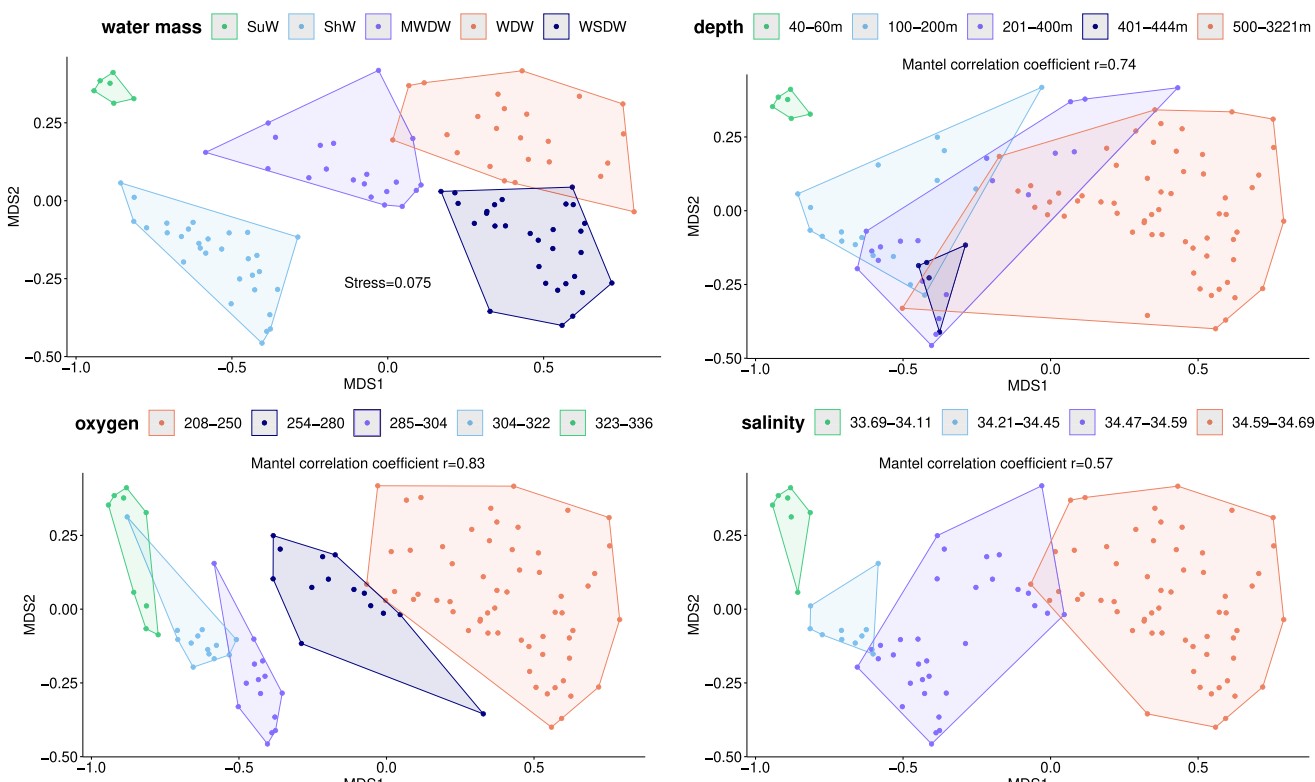

**Fig. 2 Two-dimensional NMDS plots illustrating pairwise Bray-Curtis distances between pico-nano eukaryotic communities in the Western Weddell Sea.** All NMDS plots are identical (with a stress value = 0.075 shown in an upper left panel), but the samples are clustered in different ways as described below. The water masses are mapped on the NMDS plot in the upper left corner along with three most important abiotic variables (see the other three plots): depth, oxygen concentration, and salinity. Boundary points on gradients of those abiotic variables (where pico-nano eukaryotic community composition is significantly different across the boundary) were identified using a split moving-window analysis of ecological differentiation based on a Z-score cutoff of 1, and samples were clustered according to these boundary points. The clusters are colored and marked with polygons. The intervals of the abiotic variable corresponding to the clusters are shown in the legends on top of each plot. Mantel correlation coefficients (calculated for the matrix of Bray-Curtis distances between the pico-nano eukaryotic communities and for a matrix of Bray-Curtis distances between the samples based on an abiotic variable) are shown for each continuous variable. For similar mapping of the other environmental variables measured in this study see Suppl. Fig. 1. The source data underlying this figure see in Suppl. Data 1 and Suppl. Data 2.

When we examined non-linear correlations between continuous environmental variables and the relative abundance or OTU richness of these clades individually using generalized additive models (GAMs), oxygen concentration, salinity, and water depth emerged as variables explaining the highest proportion of variance in abundance or richness of specific taxonomic groups (Suppl. Data 4). Oxygen concentration has the highest proportion of explained variance among all variables tested, and also leads according to the number of clades whose abundance or richness is well predicted by the variable (Suppl. Fig. 3, Suppl. Data 4). This result is not unexpected since, as shown above, oxygen

concentration among the continuous variables best reflects the subdivision of our sample set into water masses. Thus, from our results it appears that pico-nano eukaryotic communities change along with water masses in the Weddell Sea as they move after leaving the surface where primary production occurs. Our sample set includes only five water masses with specific movement patterns, and we do not generalize our results to the global scale.

**Abundance and diversity of pico-nano eukaryotic clades across water masses and along the oxygen and depth gradients.** To

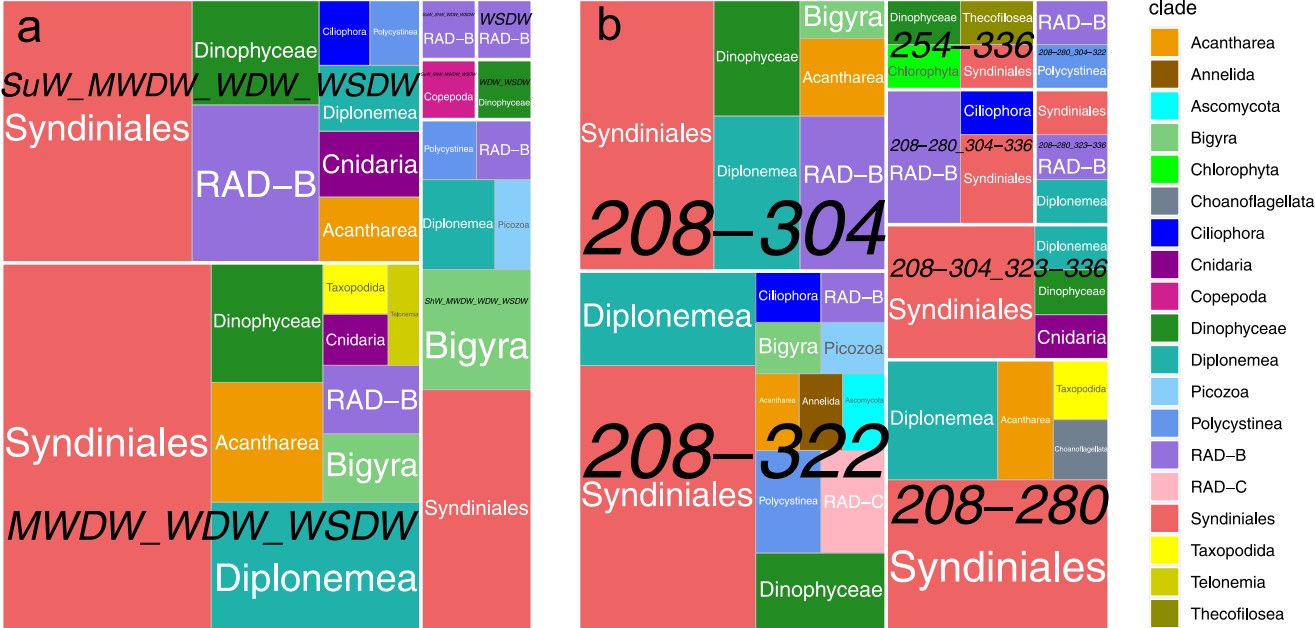

**Fig. 3 Square pie charts illustrating taxonomy of OTUs leading according to the Indicator Value index.** One hundred such OTUs were identified for the five water masses (panel **a**), and this analysis was repeated for the five intervals of oxygen concentration (panel **b**) separated by the ecological boundaries (Fig. 2). An OTU may be an indicator for a subset of water masses or oxygen concentration intervals, and such subsets are labeled in black. Taxonomic groups are labeled and color-coded according to the legend. The source data underlying this figure see in Suppl. Data 6 and Suppl. Data 7.

understand which groups of eukaryotes are specific to certain water masses or oxygen concentration ranges, we identified "indicator OTUs" (Suppl. Data 5, Suppl. Fig. 4) using a methodology described by de Cáceres and Legendre[28,29] (see "Methods"). The relative abundance of indicator OTUs differs significantly among sample categories (or groups of categories), e.g., among water masses or groups of water masses. In the context of our dataset, all water masses have their own specific indicator syndinians and dinophytes (see Suppl. Figure 4a). Indicator OTUs belonging to ciliates were found in SuW, ShW, and WDW. Representatives of the following taxa were detected as an indicator in a single water mass: diatoms (SuW); thecofiloseans (ShW); cnidarians and annelids (MWDW); diplonemids (WDW); polycystine radiolarians and RAD-B (WSDW). The following taxa were predominant among the indicator OTUs in the samples with the lowest oxygen concentrations in our dataset (208–250 μM): syndinians, dinophytes, diplonemids, and polycystine radiolarians (Suppl. Figure 4b). The determination of indicator OTUs was based on relative abundance of OTUs, which could change for certain OTUs, while their absolute abundance remained constant due to changing absolute abundances of other OTUs.

To overcome this drawback of our approach, we focused on 100 OTUs with the highest Indicator Value index[30] calculated for the water masses or oxygen concentration ranges corresponding to the ecological boundaries inferred above (Indicator Value >0.99, p-value <0.008). We argue that since these are the best indicator OTUs in the dataset, it is unlikely that a strong correlation of their relative abundance with the categorical abiotic variable is mediated by other OTUs in our dataset, with a weaker correlation to this abiotic variable (with lower Indicator Value indices). In other words, we believe that top indicator OTUs are probably more influenced by the abiotic variable than by the other OTUs in our dataset. MWDW, WDW, and WSDW form a group of water masses characterized by a substantial subset of these 100 best indicator OTUs: 23 syndinian, 8 diplonemid, 4 acantharean, and 4 dinophyte OTUs, among others (Fig. 3a).

These three water masses and SuW are characterized by 15 syndinian, 6 RAD-B, and 4 dinophyte OTUs, among others (Fig. 3a). Similar results for the oxygen concentration ranges are shown in Fig. 3b, and syndinians, diplonemids, and dinophytes lead according to the number of indicator OTUs.

In summary, there is a community of eukaryotes characteristic for deep water environments where oxygen has been partly depleted: syndinians, dinophytes, diplonemids, acanthareans, polycystines, and RAD-B radiolarians. These groups are notable in the set of best indicator OTUs (Fig. 3), and their relative abundance and OTU richness are most dependent on oxygen concentration (except for dinophytes), as compared to the other clades (Suppl. Fig. 3). Environmental distribution of this community in the Western Weddell Sea is illustrated in greater detail in Fig. 4. For each of the six taxonomic groups, we show relative abundance or OTU richness across the five water masses (Fig. 4a, b) and dependence of these metrics on depth (Fig. 4c, d) or oxygen concentration (Fig. 4e, f). Distributions of all 26 clades are shown across water masses and along the depth and oxygen concentration gradients in Suppl. Figs. 5 and 6, respectively. Relative abundance and especially richness of the six deep-water clades follow similar trends (Fig. 4), as supported by clade vs. clade Spearman correlation coefficients for relative abundance or OTU richness (Suppl. Fig. 7). OTU richness of diplonemids, syndinians, acanthareans and polycystines are especially tightly correlated in our dataset, with Spearman ρ above 0.88 (Suppl. Fig. 7). Interestingly, the OTU richness metric is expected to depend less on that of other clades as compared to relative abundance (since the latter metric is normalized on the total eukaryotic read count). Thus, the strong correlation of OTU richness for these four groups probably reflects real ecological interactions in a trophic network.

## Discussion
For the mesopelagic and bathypelagic layers, the origin of a sample from a given water mass and its subsequent fate were shown to be the most important factors affecting the community

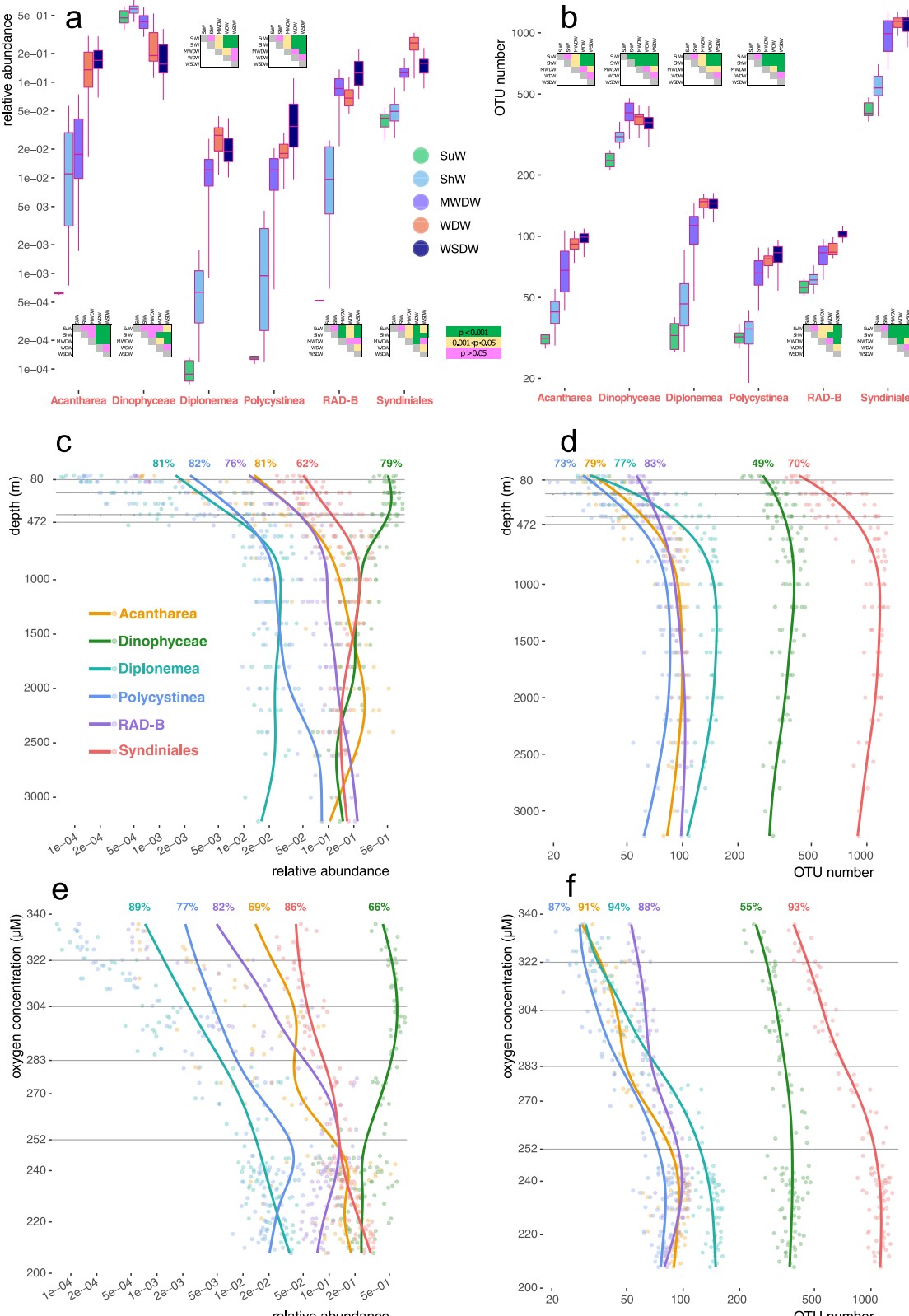

composition of both prokaryotic[31–35] and eukaryotic microorganisms,[9,14]. For example, Wilkins et al.[35] showed that advection (transport by water masses such as CDW, AABW, Antarctic Intermediate Water, and Subantarctic Mode Water) shapes prokaryotic communities in the Southern Ocean, controlling for effect of environmental variables and distance.

Pernice et al.[9] conducted a global sampling in the bathypelagic zone mainly between 30°N and 30°S and reported metabarcoding (V4 region of the 18S rRNA gene) and metagenomic data for protists; NADW, WSDW, CDW, and their mixtures were sampled. Ascomycota and Basidiomycota were abundant in the bathypelagic waters of the Pacific and Indian Oceans (in CDW-

**Fig. 4 Environmental distribution of six key players in the deep-sea eukaryotic community.** Relative abundance values and OTU counts for six taxonomic groups are shown across the five water masses sampled in this study, along with results of relevant dispersion analyses (panels **a**, **b**): either multiple ANOVA or the non-parametric Kruskal-Wallis test followed by post-hoc Dunn's test, depending on a preceding test for normality (Suppl. Fig. 8). The statistical tests were applied to sample sets of the following sizes: 6 samples (Surface Water), 30 samples (Shelf Water), 20 samples (Modified Warm Deep Water), 24 samples (Warm Deep Water), 30 samples (Weddell Sea Deep Water). We also follow relative abundance or OTU richness of these six taxonomic groups along the depth (**c**, **d**) and oxygen concentration (**e**, **f**) gradients. The y-axis scales are logarithmic in panels (**a**) and (**b**), and the x-axis scales are logarithmic in panels (**c**), (**d**), (**e**), and (**f**). Similar results for all the 26 clades explored in this study are shown in Suppl. Figs. 5 and 6. Trend lines calculated using the GAM approach are shown, along with percentage of variance explained and individual data points. Depths and oxygen concentration values identified as ecological boundary points (Fig. 2) are marked with horizontal gray lines. The source data underlying this figure see in Suppl. Data 1.

enriched samples or in mixtures of the CDW and WSDW masses), whereas they were scarce elsewhere (in pure CDW and in NADW samples). In line with this study, fungi represent a minor component of deep-sea protist communities in the Weddell Sea. Polycystines (spummelarians and collodarians) were predominant in most CDW and NADW samples, chrysophytes were prominent in NADW and in northernmost CDW samples, and syndinians (mostly MALV-II) were widespread in all water masses[9]. Diplonemids were hardly detected in that study because their V4 rDNA region is generally longer[36] than the 600 bp limit of the protocol used by Pernice et al.[9].

Zoccarato et al.[14] studied protist communities (in the 2–200 μm size fraction) in thirteen samples from four Ross Sea water masses: from newly formed HSSW and Ice Shelf Water (ISW) with a high oxygen concentration and high prokaryotic abundance, from older AABW, and the oldest CDW with a low oxygen concentration and a high nitrate and silicate content. Zoccarato et al. generated a set of ~114,000 V9 rDNA reads for ~1700 OTUs. Overall, our results from the Weddell Sea, i.e., the relative abundance and OTU richness of predominant protist clades, are consistent with the results from the Ross Sea, where strong stratification of communities by water masses was also observed[14]. Of the four water masses studied by Zoccarato et al., CDW demonstrated the highest relative abundance and richness of radiolarians (polycystines, mainly collodarians) and excavates (diplonemids). Dinoflagellates (dinophytes and syndinians) emerged as the most abundant and OTU-rich clade in all four water masses. In general, a unique protist community was found in CDW as compared to much "younger" water masses (AABW, HSSW, ISW). This result is in line with our finding that the deep-sea protist community composed of syndinians, dinophytes, diplonemids, acanthareans, polycystines, and RAD-B radiolarians is most prominent in "old" water masses such as CDW or WDW that have spent a long time in the deep ocean.

Our study extends these two key studies summarized above [9,14] by reporting data from an unsampled geographic region and unsampled water masses. It also substantially increases coverage of metabarcoding data, provides a more detailed and updated taxonomic annotation, and thus allows more accurate characterization of the protist community typical for "old" water masses which left the surface a long time ago. For example, we improve our understanding of the ecology of diplonemids, an important component of this deep-sea community. On the other hand, our results broadly confirm previous studies[9,14] regarding the composition of this protist community and confirm the finding that the movement and mixing of different water masses is the most important factor shaping protist communities in the bathypelagic and mesopelagic zones. Below we discuss the key players in the deep-sea protist community one by one.

The abundance and especially the richness of Dinophyceae correlates poorly with the five other taxonomic groups characteristic to deep water masses (Suppl. Fig. 7). Dinophyceae is the most abundant taxonomic group in the photic zone in our dataset, accounting for more than 50% of all eukaryotic reads at a depth of around 200 m (Fig. 4c). These alveolates comprise both

photo- and heterotrophic taxa, and the short V9 barcode does not allow for more precise taxonomic assignment. Their relative abundance peaked at the surface, then declined steeply, only to rise up towards the deepest part of the water column at around 2600–3200 m, where they accounted for ~8% of all eukaryotic reads. Mixotrophic dinophytes were found to be very efficient bacterial grazers even during polar summer, when their phototrophic lifestyle should prevail[17].

In addition to living cells, DNA is also preserved in dead biomass and in an extracellular state in water[37]. On the other hand, turnover of RNA in the environment is much faster, and RNA therefore more realistically reflects the presence of metabolically active organisms at the time of sampling. Giner et al.[7] proposed using changes in the RNA-to-DNA ratio for a given taxonomic group as a proxy for changes in metabolic activity. Our sampling design did not allow us to distinguish between dead and live mixotrophic algal biomass, but given the high frequency of mixotrophy among dinophytes[38,39] we hypothesize that deep-sea (e.g., mesopelagic) dinophytes are metabolically active in the Weddell Sea, as shown by Giner et al.[7] for other regions of the ocean, using RNA:DNA ratios.

RAD-B radiolarians and acanthareans are most metabolically active in the mesopelagic zone, whereas polycystines only at the surface[7]. Thus, relatively abundant polycystine radiolarians in the bathypelagic zone could be explained, at least in part, by dead cells. In addition, radiolarians were shown to be responsible for remarkable carbon export events and likely form a major component of the gelatinous material collected in sediment traps[40–42]. In marine plankton, polycystines are represented mainly by orders Spumellarida and Collodaria[4,9]. The latter form large colonies, harbor photosymbionts, and are reported from the surface during the summer blooms[43]. On the other hand, some spumellarids also inhabit the meso- and bathypelagic depths[43]. Therefore, the progressive increase in the relative abundance of polycystines may be caused by a cumulative effect of sinking dead collodarian biomass and a deeper spumellarid activity maximum. Acanthareans are a locally very abundant lineage of Radiolaria with largely unknown biology[43]. Some form large fast-sinking cysts (a metabolically active reproductive stage)[44], which are likely responsible for the activity peak observed for acanthareans in the mesopelagic zone[7].

We also expand knowledge of one of the most overlooked heterotrophic protists in marine aphotic plankton: an abundant and extremely OTU-rich group called diplonemids[6,36,45]. Taking advantage of our homogeneous dataset and high-resolution depth transects, we refined the existing distribution curves for diplonemids[46] across depth and oxygen gradients. We also showed that the relative abundances of syndinians and diplonemids and their OTU counts are tightly correlated (Spearman $\rho = 0.83$, 0.88, respectively; Suppl. Fig. 7), as indicated by their similar OTU count and relative abundance curves (Fig. 4). In fact, the correlation between these groups is among the strongest ones observed for eukaryotic phyla in our study (Suppl. Fig. 7). Published RNA-DNA comparisons[7,47] revealed that syndinians and excavates (comprised almost entirely of diplonemids in similar

V9 metabarcoding datasets)[46] reach their highest metabolic activity in the mesopelagic zone, consistent with the relative abundance peaks observed in our study. Boeuf et al.[40] reported much lower RNA-DNA ratios for both groups at a depth of 4000 m, but the ratio for excavates was an order of magnitude higher than that for syndinians. This again agrees with observed decrease in relative abundance of both groups throughout the bathypelagic zone reported here and with recently published data on the benthic diversity[6].

Syndinians (basal dinoflagellate lineages also known as Marine Alveolates or MALVs) are globally abundant and diverse parasitoids of various planktonic lineages[1,48,49]. While we know nothing about the biology of 99% of diplonemid diversity[45], some representatives were described as putative parasitoids[50] as well as bacteriovores[51,52]. The above-described patterns suggest that diplonemids are linked to syndinians either through direct trophic interactions or by utilizing the same food sources, possibly radiolarians also abundant in their mesopelagic habitat. Given our finding that the deep-sea protist community lives in "old" oxygen-depleted water masses, we suggest that attempts at cultivating these protists should be done at low oxygen concentrations. Protists such as deep-sea diplonemids and radiolarians have so far resisted all cultivation attempts.

The existence of distinct heterotrophic protist communities in the photic and non-photic layers of the water column was previously described on the global scale[6–8]. Moreover, it was demonstrated that the mesopelagic zone is a hotspot of protist metabolic activity[7], and that corresponds to the fact that about 90% of carbon is respired in this layer[53]. However, here we present the most detailed picture of the vertical distribution of protist clades from the Southern Ocean and refine the description of protist distribution by moving from vertical depth stratification to ecological succession within moving water masses.

Protist communities in the polar oceans were shown to be distinct from each other and from the tropical/temperate cosmopolitan community. This was demonstrated in the photic zone[12] as well as in the benthos[6]. Thus, it is important to expand sampling in the Southern Ocean to obtain a truly global picture of protist diversity and biogeography.

## Methods

**Plankton sampling.** This study is based on marine planktonic samples collected in the Weddell Sea from February to April 2019 during the PS118 expedition of the research vessel Polarstern (Alfred Wegener Institute for Polar and Marine Research, Bremerhaven, Germany). The water samples were taken by 12-liter Niskin bottles attached to a carousel (SBE32, SN 718) equipped with a CTD (SN 937). The system included two sensor pairs for conductivity (SBE4, SN 3590, SN 3570) and temperature (SBE3, SN 5112, SN 5115), one high-precision pressure sensor Digiquartz 410K-134 (SN 937), one oxygen sensor (SBE43, SN 1834), one transmissometer (Wetlab C-Star, SN 1198), one fluorometer (Wetlab FLRTD, SN 1853), and one altimeter (Benthos PSA-916, SN 47768). Information about samples including depth, geographical coordinates, fluorescence intensity, oxygen concentration, salinity, and water temperature are listed in Suppl. Data 1. Storage of water samples awaiting filtration was carried out at 0 °C in the dark, and filtration of 10 liters of seawater took 2 to 4 h and was carried out at +4 °C in the dark. Seawater (10 liters) was filtered using a Cole-Palmer Masterflex benchtop peristaltic pump through a system of three filters: a 180 μm nylon mesh (diameter 47 mm, Isopore, Ireland), 20 μm nylon mesh (diameter 47 mm, Isopore, Ireland), and a 0.8 μm polycarbonate filter (diameter 47 mm, Isopore, Ireland). Pico-nano marine plankton (the 0.8–20 μm size fraction) was collected in this way and kept on filters at −20 °C in 1 ml of the PW1 lysis buffer taken from the PowerWater DNA Isolation Kit (MO BIO, USA).

**DNA isolation, sequencing, and data processing.** DNA was isolated from the filters using the PowerWater DNA Isolation Kit (MO BIO, USA). The V9 region of the 18S rRNA gene was amplified using universal eukaryotic primers[54] (1389F 5'-TTGTACACACCGCCC-3', 1510R 5'-CCTTCYGCAGGTTCACCTAC-3'; 94 °C for 15 s, 62 °C for 30 s, 68 °C for 30 s, 30 cycles). V9 amplicon barcoding

libraries were sequenced using an Illumina HiSeq 4000 instrument at Genome Quebec, and paired-end 150 bp reads were generated. Primer sequences were removed from reads using *cutadapt* v. 1.15 under the following settings: --no-indels, --discard-untrimmed, --minimumlength 50, --overlap 4, -e 0.2, -a TTGTACACACCGCCC… GTAGGTGAACCTGCRGAAGG, -A CCTTCYG-CAGGTTCACCTAC… GGGCGGTGTGTACAA. Reads were then merged using *bbmerge* under the default settings. Using *bbduk*, merged reads containing undetermined bases (Ns) were filtered out and reads having average Phred quality below 20 were discarded. Cleaned reads were dereplicated into barcodes using *vsearch* v. 2.7.1 under the default settings. Barcodes were grouped into OTUs using *swarm* v. 2.2.2[55] under the following settings: -d 1, -f, -z. OTUs were taxonomically annotated using the *ggsearch36* tool from the FASTA package (ftp://ftp.ebi.ac.uk/pub/software/unix/fasta) under the following settings: -m 8 -d 0 -b 1 -E 10 -T 40 -w 199. As references for the annotation we used rRNA sequences from the PR2 database (https://github.com/pr2database/pr2database) supplemented by updated annotations for Discoba and Metamonada[56] taken from the EukRef database (https://github.com/eukref/curation). The V9 region was extracted before the taxonomic assignment. We inspected the distribution of OTUs across abundance bins and the distribution of OTUs according to similarity to reference sequences and decided to exclude from downstream analyses OTUs having abundance below 100 reads and similarity to reference below 90%. At this step, 146.4 million reads (95.9% of the original reads) and 5,048 OTUs (10.8% of the original OTUs) were retained. Finally, we selected 26 eukaryotic clades (defined according to Adl et al.[57]) represented by at least 200,000 reads. These clades account for 94.6% of the original reads and 10.1% of the original OTUs.

**Statistics and reproducibility.** All statistical analyses and plotting were performed using R v. 4.2.1. Bray-Curtis distances between eukaryotic communities were calculated using the *decostand* and *metaMDS* functions of the *vegan* package v.2.6-4. For NMDS analysis, we first explored as input either a Hellinger-transformed (using the *decostand* function) or non-transformed relative abundance matrix and tested dimensionalities from to 2 to 6 (using the *metaMDS* function, with 50 to 1000 iterations). Judging by the stress values across a range of dimensionalities, we chose Hellinger-transformed data and 2 dimensions (stress = 0.075). We also performed a split moving-window analysis of ecological differentiation[58] that located significant changes in community composition across a gradient of an environmental variable (sample groups were defined based on an absolute Z-score cutoff of 1 and were visualized on the NMDS plot). For this analysis, we used the *smwda* function of the *EcolUtils* v.0.1 package (window size = 10, 1000 replicates). The Bray-Curtis distance matrix based on Hellinger-transformed relative abundance values was used as an input.

The Mantel test was used to calculate correlation coefficients for the matrix of Bray-Curtis distances between the pico-nano eukaryotic communities (based on Hellinger-transformed relative abundance values) vs. a matrix of Bray-Curtis distances between the samples based on an abiotic variable (Euclidean distances were used in the case of geographic coordinates). The *mantel* function of the *vegan* package v.2.6-4 was used (the Pearson correlation coefficient was used, and 10,000 permutations were performed).

The distributions of depth, oxygen concentration, salinity, temperature, and relative abundance values as well as OTU counts of six key players in the deep-sea eukaryotic community (Acantharea, Dinophyceae, Diplonemea, Polycystinea, RAD-B, and Syndiniales) over five water masses defined by water mass theory were tested for normality by creating Quantile-Quantile plots (with the *ggqqplot* function of the *ggpubr* package v.0.4.0) and performing the Shapiro–Wilk test. If the Quantile-Quantile plots corresponded to a normal distribution and the *p*-value of the Shapiro-Wilk test exceeded 0.05, we considered the distributions to be normal and compared their variances using multiple ANOVA followed by the Tukey correction. Otherwise, we compared their variances using the non-parametric Kruskal–Wallis test followed by post-hoc Dunn's test (the *dunnTest* function of the *FSA* package v.0.9.3, method = "bonferroni").

Distributions of biological variables (relative abundance and OTU richness) versus environmental variables (depth, oxygen concentration, water temperature, salinity, fluorescence intensity) were approximated using generalized additive models (GAM) as implemented in the *mgcv* package v.1.8-38. We fitted GAMs (y ~ s(x)) based on the beta distribution that is suitable for variables between 0 and 1 (relative abundance) or GAMs based on the gamma distribution that is suitable for positive continuous variables (OTU richness). In the latter case, the setting 'link = identity' was used. We calculated pairwise Spearman rank correlation coefficients for relative abundance and richness of the 26 selected eukaryotic clades and applied hierarchical clustering to this matrix (the *hclust* package, method 'complete').

Indicator OTUs were identified using the *indicspecies* package v.1.7.8 implementing Indicator Values introduced by Dufrene and Legendre[30] and protocols introduced by De Cáceres and Legendre[28,29]. The matrix of Bray-Curtis distances between the pico-nano eukaryotic communities (relying on the non-transformed relative abundance values) was used as an input. We used the *multipatt* function with 10,000 permutations. Indicator OTUs were inferred for the water masses (and sets of several water masses) or for ranges of oxygen concentration (and sets of ranges) defined by the ecological boundary approach.

**Reporting summary**. Further information on research design is available in the Nature Portfolio Reporting Summary linked to this article.

## Data availability

Raw metabarcoding reads generated in this study are available at NCBI under a BioProject accession number PRJNA783769. Input data for the figures are available as spreadsheets in Supplementary Data files, and all other data generated in this project are available upon request from the corresponding author.

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

## Acknowledgements

We thank the Alfred Wegener Institute (AWI) and the R/V Polarstern crew and captain for their support (AWI_PS118_13). We especially thank Markus Janout (AWI) for sharing physical oceanography data and Boris Dorschel (AWI) for organizing the cruise. We acknowledge the expertise and services of Genome Quebec, where the amplicon sequencing took place, and the support from the Czech Science Foundation (project 18-23787S to A.H.), from the Czech Ministry of Education, Youth, and Sports (project ERC CZ LL1601 to J.L.) and the Gordon and Betty Moore Foundation (GBMF #9354 to J.L.).

## Author contributions

A.H., P.F., O.F., and J.L. designed the project. A.H. and J.L. provided the funding. O.F. and N.J. generated and analyzed the data. O.F., P.F., A.H., and J.L. took part in writing the manuscript and responded to reviewers' comments.

## Competing interests

The authors declare no competing interests.
