## [Peer Review File · Communications Biology]

Reviewers' comments:

Reviewer #1 (Remarks to the Author):

The manuscript by Olga Flegontova and colleagues describes a study aimed at characterizing the diversity of small (<20 µm) protists within defined water masses (WM) in the NW Weddell Sea. The sampling design, the procedures and methods utilized are appropriate to fulfil the aims of the study as well as most of the statistical analyses. The outcomes expand the current knowledge on protist biodiversity to the Weddell Sea area for the first time and will be of interest not only to biological oceanographers. The article is clear and in general straightforward. I could not detect any major issue in it, although some wrinkles need to be flattened before I can fully endorse its publication.

Although I understand that no strict formatting style is required at this stage, I had some difficulties in following the methodological workflow, being a Method section absent in the current version. I strongly recommend the authors to reorganize all the information that are spread here and there in the manuscript (and supplement) so that a reader can easily look for (and find) the needed details. I also suggest reporting the brand and the model of the multiparametric probe (supplement, line 5) and to describe in a clearer way (it is understandable, but the style can be improved) the seawater filtration steps (changing µl into µm where appropriate – supplement lines 9-14).

L24-25: I would remove 'during a Polarstern Antarctic cruise' from the abstract

L 39: ...abundance exceed that of EUKARYOTIC PRIMARY producers...

L 67: please check the format of references in the text

L 104 onward: in the Results section a quick description of the overall composition of the eukaryotic communities should be provided so that the reader can have clearer idea of the relative abundance of the different taxa (bar plots usually serve this scope). In fact, Figure 3 is based on the outcomes of the indicator OUT analysis and does not report the 'original' data. The study is based on an impressive number of samples and it would be a pity non to show the data... maybe as averages of the taxa relative abundance within the different water masses?

L 113: what is the rationale for including in the analysis the 'depth of the seafloor' (often referred to as 'elevation'... in most studies called bottom depth)?

L 127: do the data meet the requirements for running an ANOVA? I might be wrong but having a look at Figs 1B and 1D it does not seem that depth and oxygen concentration datasets follow a normal distribution. Please check.

L 127-131: I encourage the authors to re-write the sentences. The outcomes of ANOVA highlight dissolved oxygen as the variable that mostly discriminate the samples within the water masses. I would be cautious in calling it a proxy. By definition, the combination of potential temperature and salinity data defines water masses. The wording used in lines 142-143 '...most closely reflect the subdivision of our sample set into WM...' is more appropriate. Moreover, it is well known that oxygen concentration is dependent on the 'age' of the WM, therefore it is not strictly a belief of the authors.

L 144-146: The authors convinced me about the fact that in different WMs, different eukaryotic communities reside. However it is not only a matter of communities' evolution (it is not an evolution, but rather a change, a modification) along with the flowing of WMs into the ocean's interior, it is also a matter of mixing. In fact MWDW are formed by the mixing of SuW and WDW (lines 77-78), therefore I suggest to include the mixing among the forces driving the observed differences among community structures.

L153-164: even if I understand the meaning of this paragraph I suggest to re-write it for clarity and in a more fluent (less bullet-point-like) style.

L 186: In Figure 4C, D, E, and F I would switch X and Y axes for a more intuitive understanding (highest depths at the bottom of the plots)

L 242: ...the movement AND MIXING of distinct water masses...

L 251: please consider changing 'spectrum' into 'water column'

L 254-266. These paragraphs have a high degree of speculation and do not add much to the merit of the article. The present study is based on DNA and in my opinion there is no need to include caveats addresses to the activity of cells...

Reviewer #2 (Remarks to the Author):

The manuscript of Flegontova and colleagues is a compelling study of pelagic protists along the water column and among several water masses in the Weddell Sea. It shows that water masses, and notably their oxygen content explains well the structure of protists communities. The authors compare the diversity patterns obtained in the Weddell Sea with other omics-based studies, linking their observations to broad-scale patterns as well as with other studies focusing on the Southern Ocean. They develop a convincing discussion that pushes our knowledge on the ecology of pelagic protists along the water column (and their possible trophic interactions) beyond the state of the art. Therefore, I recommend the publication of the paper, although some minor issues should be addressed (e.g. NMDS, see below).

Minor comments:

- The result section contains few elements of discussion (e.g. L.129-131; 141-146; 162-164; 192-195).
- L.159. "And the following taxonomic groups lead according to [...]" something it not right with this sentence.
- L. 166: "[...] we focused on 100 OTUs leading [...]" again the lead term that is I think unclear. Maybe you mean with best indicator value?
- L.198. meso and bathypelagic *layers* instead of *regions*.
- L.199. "history"? what do you mean?
- - L.226. *More so* instead of *Moreso*

Additional comments on methods (supplementary)

- L.17. lease add the reference of the chosen primers.
- L.18. Could you please add some details on the library prep for sequencing?
- L.25. What do you mean by "Cleaned reads were collapsed into barcodes [...]" ? Do you mean dereplicated reads?
- L.27. Any reference for ggsearch36? Also, you may give some details on the taxonomic annotation (cutoff, taxonomic consensus?). As it is, there is no information given.
- L.37. You specify that you did you NMDS ordination on four dimensions, since it gave the lowest stress value (0.038). However, figure 2 shows only the two first dimensions, and you still report that the stress value is 0.038 for those ordinations. The stress value of a NMDS on four dimensions cannot be used for two-dimensions ordinations. I suggest redoing the NMDS on only two dimensions. Even if the stress value is higher, it will better represent the compositional structure of the data.

Reviewer #3 (Remarks to the Author):

This is a review report for the manuscript, entitled "Water masses shape pico-nano eukaryotic communities of the Weddell Sea" (Manuscript Number: COMMSBIO-22-2925). This study focuses on factors affecting the distribution of planktonic pico-nano eukaryotes and observe an ecological succession of eukaryotic communities as the water masses move away from the surface and as oxygen becomes depleted with time of the Weddell Sea. However, the experimental design did not take into account the effect of nutrient (N, P, Si) on the community and did not classify the species into nutrient metabolism types for analysis. The logic of the writing in the results section is lacking, and there is no focus on the effect of water masses on the community. My opinion is not suitable for publication in this journal.

Below are some specific comments that I hope will be helpful to the manuscript.

1. Method: "Filtration and storage of water samples awaiting filtration was carried out at +4°C in the dark, and filtration of 10 liters of seawater took 2 to 4 hours." Whether the surface temperature in the investigated sea area is below 0°C and the filtration time at +4°C for 4 hours has an effect on the samples?
2. Method: "... 20 µl nylon mesh (diameter 47 mm, Isopore, Ireland), and a 0.8 µl polycarbonate filter (diameter 47mm, Isopore, Ireland). Pico-nano marine plankton (0.8-20 µl size fraction)....". 20 µl? 0.8 µl? 0.8-20 µl?
3. Figure 1A: Labeled with the name of the land.
4. Page 4: "We believe this is not a coincidence since oxygen concentration is likely correlated with the time that passed after the last contact of the water with the surface." I don't think so.
5. What are the characteristics about the five water masses with specific movement patterns. In turn, we analyze those characteristics in different water masses that lead to differences in the communities.
6. Results: Suggested additions: effects of different characteristic water masses on communities, variability of vertical water depth communities.
7. The discussion section is modified in conjunction with the content of the results.

Reviewer #1 (Remarks to the Author):

The manuscript by Olga Flegontova and colleagues describes a study aimed at characterizing the diversity of small (<20 µm) protists within defined water masses (WM) in the NW Weddell Sea. The sampling design, the procedures and methods utilized are appropriate to fulfil the aims of the study as well as most of the statistical analyses. The outcomes expand the current knowledge on protist biodiversity to the Weddell Sea area for the first time and will be of interest not only to biological oceanographers. The article is clear and in general straightforward. I could not detect any major issue in it, although some wrinkles need to be flattened before I can fully endorse its publication.

Although I understand that no strict formatting style is required at this stage, I had some difficulties in following the methodological workflow, being a Method section absent in the current version. I strongly recommend the authors to reorganize all the information that are spread here and there in the manuscript (and supplement) so that a reader can easily look for (and find) the needed details.

A: We have reorganized the paper as suggested, moving the Methods section to the main text.

I also suggest reporting the brand and the model of the multiparametric probe (supplement, line 5) and to describe in a clearer way (it is understandable, but the style can be improved) the seawater filtration steps (changing µl into µm where appropriate – supplement lines 9-14).

A: We have added model numbers for all the measurement instruments and added few details about the water sampling. The typo (µl into µm) was also corrected.

L24-25: I would remove 'during a Polarstern Antarctic cruise' from the abstract

A: corrected as requested.

L 39: ...abundance exceed that of EUKARYOTIC PRIMARY producers...

A: rephrased.

L 67: please check the format of references in the text

A: We have corrected the references as requested.

L 104 onward: in the Results section a quick description of the overall composition of the eukaryotic communities should be provided so that the reader can have clearer idea of the relative abundance of the different taxa (bar plots usually serve this scope). In fact, Figure 3 is based on the outcomes of the indicator OTU analysis and does not report the 'original' data. The study is based on an impressive number of samples and it would be a pity non to show the data... maybe as averages of the taxa relative abundance within the different water masses?

A: We have added figures as requested. Relative abundance values and OTU numbers are shown in the form of boxplots for six most abundant taxa in Fig. 4A,B, and for all taxa in Supp. Fig. 5. These values are also shown in the form of GAM trendlines for all taxa along the depth and oxygen gradients (Suppl. Fig. 6).

L 113: what is the rationale for including in the analysis the 'depth of the seafloor' (often referred to as 'elevation'... in most studies called bottom depth)?

A: Some protists such as *Diplonema* of the Diplonemea phylum 'prefer' coastal areas, and for this reason we included this variable, although in our dataset it has very little influence on protist communities. We renamed this variable: from 'depth of the seafloor' to 'bottom depth'.

L 127: do the data meet the requirements for running an ANOVA? I might be wrong but having a look at Figs 1B and 1D it does not seem that depth and oxygen concentration datasets follow a normal distribution. Please check.

A: We have tested distributions of all the variables for normality (see Suppl. Figs. 2 and 8), and for those failing the test (depth, salinity, and temperature) we have replaced ANOVA by the non-parametric Kruskal-Wallis test followed by post-hoc Dunn's test. Interestingly, oxygen concentration was not among them. Figs. 1 and 4 were updated accordingly.

L 127-131: I encourage the authors to re-write the sentences. The outcomes of ANOVA highlight dissolved oxygen as the variable that mostly discriminate the samples within the water masses. I would be cautious in calling it a proxy. By definition, the combination of potential temperature and salinity data defines water masses. The wording used in lines 142-143 '...most closely reflect the subdivision of our sample set into WM...' is more appropriate. Moreover, it is well known that oxygen concentration is dependent on the 'age' of the WM, therefore it is not strictly a belief of the authors.

A: We have rephrased the text according to the reviewer's suggestion and added three references supporting the statement that oxygen concentration is dependent on the 'age' of water masses: Hoppema et al. 1997, Karstensen and Tomczak 1998, Vernet et al. 2019.

L 144-146: The authors convinced me about the fact that in different WMs, different eukaryotic communities reside. However it is not only a matter of communities' evolution (it is not an evolution, but rather a change, a modification) along with the flowing of WMs into the ocean's interior, it is also a matter of mixing. In fact MWDW are formed by the mixing of SuW and WDW (lines 77-78), therefore I suggest to include the mixing among the forces driving the observed differences among community structures.

A: We have rephrased the text according to the reviewer's suggestion.

L153-164: even if I understand the meaning of this paragraph I suggest to re-write it for clarity and in a more fluent (less bullet-point-like) style.

A: We have rephrased the text so that it is more fluent and clearer.

L 186: In Figure 4C, D, E, and F I would switch X and Y axes for a more intuitive understanding (highest depths at the bottom of the plots)

A: We have followed reviewer's suggestion and edited Figure 4 accordingly.

L 242: ...the movement AND MIXING of distinct water masses...

A: We have changed according to the reviewer's suggestion.

L 251: please consider changing 'spectrum' into 'water column'

A: We have changed according to the reviewer's suggestion.

L 254-266. These paragraphs have a high degree of speculation and do not add much to the merit of the article. The present study is based on DNA and in my opinion there is no need to include caveats addresses to the activity of cells...

A: We agree that above-mentioned lines may be seen as speculative. However, we believe that raising hypotheses to inspire future research is legit and beneficial for the community. We also think the caveat is important to mention. However, if reviewer #1 and editor will insist on removal these particular lines, we are ready to accept the suggestion.

Reviewer #2 (Remarks to the Author):

The manuscript of Flegontova and colleagues is a compelling study of pelagic protists along the water column and among several water masses in the Weddell Sea. It shows that water masses, and notably their oxygen content explains well the structure of protists communities. The authors compare the diversity patterns obtained in the Weddell Sea with other omics-based studies, linking their observations to broad-scale patterns as well as with other studies focusing on the Southern Ocean. They develop a convincing discussion that pushes our knowledge on the ecology of pelagic protists along the water column (and their possible trophic interactions) beyond the state of the art. Therefore, I recommend the publication of the paper, although some minor issues should be addressed (e.g. NMDS, see below).

Minor comments:

- The result section contains few elements of discussion (e.g. L.129-131; 141-146; 162-164; 192-195).
- L.159. "And the following taxonomic groups lead according to [...]" something it not right with this sentence.

A: We have rephrased the sentence (and the whole section).

- L. 166: "[...] we focused on 100 OTUs leading [...]" again the lead term that is I think unclear. Maybe you mean with best indicator value?

A: We have rephrased the sentence.

- L.198. meso and bathypelagic *layers* instead of *regions*.

A: We have rephrased the sentence according to the reviewer's suggestion.

- L.199. "history"? what do you mean?

A: We have changed the wording to make the meaning clearer.

-- L.226. *More so* instead of *Moreso*

A: We have corrected the word according to the reviewer's suggestion.

Additional comments on methods (supplementary)

- L.17. please add the reference of the chosen primers.

A: We have added primer names and a citation (Zettler et al. 2009).

- L.18. Could you please add some details on the library prep for sequencing?

We have added all the details on sequencing we could get from the Genome Quebec

- L.25. What do you mean by "Cleaned reads were collapsed into barcodes [...]" ? Do you mean dereplicated reads?

A: Yes, we meant dereplicated reads, and we have rephrased accordingly.

- L.27. Any reference for ggsearch36? Also, you may give some details on the taxonomic annotation (cutoff, taxonomic consensus?). As it is, there is no information given.

A: We have added a link to the FASTA software package and listed all the ggsearch parameters we used. The taxonomic annotation procedure are now clarified in the Methods section.

- L.37. You specify that you did you NMDS ordination on four dimensions, since it gave the lowest stress value (0.038). However, figure 2 shows only the two first dimensions, and you still report that the stress value is 0.038 for those ordinations. The stress value of a NMDS on four dimensions cannot be used for two-dimensions ordinations. I suggest redoing the NMDS on only two dimensions. Even if the stress value is higher, it will better represent the compositional structure of the data.

A: We have redone the NMDS analysis on two dimension and modified all the figures accordingly.

Reviewer #3 (Remarks to the Author):

This is a review report for the manuscript, entitled "Water masses shape pico-nano eukaryotic communities of the Weddell Sea" (Manuscript Number: COMMSBIO-22-2925). This study focuses on factors affecting the distribution of planktonic pico-nano eukaryotes and observe an ecological succession of eukaryotic communities as the water masses move away from the surface and as oxygen becomes depleted with time of the Weddell Sea. However, the experimental design did not take into account the effect of nutrient (N, P, Si) on the community and did not classify the species into nutrient metabolism types for analysis. The logic of the writing in the results section is lacking, and there is no focus on the effect of water masses on the community. My opinion is not suitable for publication in this journal.

A: We agree that adding other important environmental variables would be highly relevant. Unfortunately, only the variables analyzed in the paper were measured during the expedition.

Moreover, previous global-scale studies on the Tara Oceans data (with dozens of environmental variables measured for a subset of the marine planktonic samples, including nutrient concentration) suggest that oxygen concentration and temperature are the most important variables affecting composition of the planktonic microbiome, and concentration of various nutrient has a much smaller effect (Salazar et al. 2019 “Gene Expression Changes and Community Turnover Differentially Shape the Global Ocean Metatranscriptome”, Flegontova et al. 2020 “Environmental determinants of the distribution of planktonic diplomonads and kinetoplastids in the oceans”).

We disagree that there is no focus in the manuscript on the effect of the water masses. In fact, that’s main point of our manuscript, and that is reflected in most figures.

Below are some specific comments that I hope will be helpful to the manuscript.

1. Method: “Filtration and storage of water samples awaiting filtration was carried out at +4°C in the dark, and filtration of 10 liters of seawater took 2 to 4 hours.” Whether the surface temperature in the investigated sea area is below 0°C and the filtration time at +4°C for 4 hours has an effect on the samples?

A: Since we worked with DNA, which should be relatively robust to much bigger changes in temperature, we believe the effect mentioned by reviewer is negligible.

2. Method: “... 20 µl nylon mesh (diameter 47 mm, Isopore, Ireland), and a 0.8 µl polycarbonate filter (diameter 47mm, Isopore, Ireland). Pico-nano marine plankton (0.8-20 µl size fraction)....”. 20 µl? 0.8 µl? 0.8-20 µl?

A: The typo was corrected: µl was replaced by µm. The numbers X-Y correspond to mesh size ranges.

3. Figure 1A: Labeled with the name of the land.

A: We disagree that any additional labelling of land masses is needed since our study is focused on the ocean, and the only relevant land mass name would be Antarctic Peninsula.

4. Page 4: “We believe this is not a coincidence since oxygen concentration is likely correlated with the time that passed after the last contact of the water with the surface.” I don’t think so.

A: We have cited two publications supporting our statement that oxygen concentration is influenced by the age of water masses:

Hoppema M, Fahrbach E, Schröder M. On the total carbon dioxide and oxygen signature of the circumpolar deep water in the Weddell Gyre.

Karstensen J and Tomczak M. Age determination of mixed water masses using CFC and oxygen data.

Moreover, reviewer 1 wrote: “Moreover, it is well known that oxygen concentration is dependent on the ‘age’ of the WM, therefore it is not strictly a belief of the authors.”

5. What are the characteristics about the five water masses with specific movement patterns. In turn, we analyze those characteristics in different water masses that lead to differences in the communities.

A: We believe the differences in the water masses are described in the relevant part of Introduction (line 65 and further). We are not clear about the remaining part of the comment, and we were not certain how to address it.

6. Results: Suggested additions: effects of different characteristic water masses on communities, variability of vertical water depth communities.

A: The suggestion above is unclear and therefore we were not certain how to address it.

7. The discussion section is modified in conjunction with the content of the results.

A: The suggestion above is unclear and therefore we were not certain how to address it.

REVIEWERS' COMMENTS:

Reviewer #1 (Remarks to the Author):

The authors took into account most of my previous (minor) suggestion, therefore I endorse the publication of the manuscript